# Admission Causes, Morbidity, and Outcomes in Scavenger Birds in the North of Portugal (2005–2022)

**DOI:** 10.3390/ani13132093

**Published:** 2023-06-24

**Authors:** Andreia Garcês, Isabel Pires, Roberto Sargo, Luís Sousa, Justina Prada, Filipe Silva

**Affiliations:** 1Exotic and Wild Animal Service, Veterinary Hospital of University of Trás-os-Montes e Alto Douro (UTAD), 500-801 Vila Real, Portugal; rsargo@utad.pt (R.S.); lsousa@utad.pt (L.S.); fsilva@utad.pt (F.S.); 2Centre for the Research and Technology of Agro-Environmental and Biological Sciences (CITAB), University of Trás-os-Montes e Alto Douro, 5001-801 Vila Real, Portugal; 3Animal and Veterinary Research Centre (CECAV), University of Trás-os-Montes and Alto Douro, Quinta de Prados, 5000-801 Vila Real, Portugal; ipires@utad.pt (I.P.); jprada@utad.pt (J.P.)

**Keywords:** vultures, mortality, scavenger birds, ecology, aspergillosis, endangered specie

## Abstract

**Simple Summary:**

This study aims to investigate the admission causes, outcomes, and mortality of vultures admitted to a wildlife rehabilitation centre and necropsy service in Northern Portugal. The data were obtained from the Rehabilitation of Wild Fauna archives centre of the University of Trás-os-Montes and Alto Douro, Vila Real, Portugal. Over 17 years (2005–2022), 84 animals were admitted: 10 *A. monachus*, 69 *G. fulvus*, and 5 *N. percnopterus*. The main causes of admission to the centre were 80% (*n* = 63) unknown cause, 13% (*n* = 10) found debilitated, 6% (*n* = 5) vehicle collision, 4% (*n* = 3) captivity, 1% (*n* = 1) gunshot, and 1% (*n* = 1) electrocution. Most animals were admitted during the summer (45.2%) and autumn (36.9%). Analysis of outcome data showed that 73% (*n* = 58) of the animals that arrived alive at the centre could be rehabilitated and released back into the wild. Thirteen animals died during treatment and five were found dead.

**Abstract:**

Portugal is the habitat of three species of vultures. According to the IUCN Red List of Threatened Species, *Neophron percnopterus* is an Endangered species, *Aegypius monachus* is nearly Threatened, and *Gyps fulvus* is of Least Concern. This study aims to investigate the admission causes, morbidity, and outcomes of vultures admitted to a wildlife rehabilitation centre and necropsy service in Northern Portugal. Over 17 years (2005–2022), 84 animals were admitted: 10 *A. monachus*, 69 *G. fulvus*, and 5 *N. percnopterus*. The main causes of admission to the centre were 80% (*n* = 63) unknown cause, 13% (*n* = 10) found debilitated, 6 % (*n* = 5) vehicle collision, 4% (*n* = 3) captivity, 1% (*n* = 1) gunshot, and 1% (*n* = 1) electrocution. Most animals were admitted during the summer (45.2%) and autumn (36.9%). Analysis of outcome data showed that 73% (*n* = 58) of the animals that arrived alive at the centre could be rehabilitated and released back into the wild. Thirteen animals died during treatment and five were found dead. This is the first time that such a lengthy study of results and mortality has been carried out for these species in Portugal. Although the data are limited, they can already provide some information about these populations, particularly for the endangered species that are so rare to observe.

## 1. Introduction

Vultures are obligate scavenging birds whose diet is based on dead animals [1]. Worldwide, 23 species of vultures are divided into 2 great groups: New-World vultures (Cathartidae) and Old-World vultures (Accipitridae). The 16 species of Accipitridae can be found in Africa, Asia, and Europe, and the 7 species of Cathartidae can be found in North and South America [2,3].

Portugal is the habitat of three species of Old-World vultures: the Egyptian vulture (*Neophron percnopterus*), the Cinereous vulture (*Aegypius monachus*), and the Griffon vulture (*Gyps fulvus*) [4]. According to the IUCN Red List of Threatened Species, *N. percnopterus* [5] is an Endangered species, *A. Monachus* [6] is nearly Threatened, and *G. fulvus* [7] is of Least Concern. These three species have enormous ranges that span three continents—Europe, Africa, and Asia [2].

The Iberian population of *A. monachus* is quite isolated from populations in eastern Europe. In Portugal, they can be found in a narrow strip in the border area of Alentejo and Beira Baixa (range 30–40 km inland). In the years 1930–1950, the colonies that existed in the Alentejo disappeared due to due to human disturbance and landscape changes (e.g., the construction of dams and the intensification of agriculture). In recent years, the number of individuals in Portugal has been increasing, with animals coming from Spain, and it is believed that it is approaching one hundred individuals [4]. Most of the breeding population in Portugal of *G. fulvus* can be found mainly in the upper Douro and Tagus valleys and their tributaries; some pairs were observed in the region of Vila Velha de Ródão, Proença-a-Nova, Serra de Penha Garcia, and Serra of S. Mamede. The population has been increasing in recent years in Portuguese territory. In Europe, there have been some population declines in Türkiye and the Caucasus [4]. In Portugal in the past, *N. percnopterus* was quite common and well distributed all over the country. From the 19th century onwards, the population declined, and it is currently found only on the borders of the Centre and Northeast of Portugal. According to the census carried out in 2000, 83 to 84 couples were counted in Portuguese territory. The population has been slowly increasing due to conservation efforts. In Europe, the population is also in general decline [4].

Their diet consists of the carcasses of animals that die from the most diverse causes [3]. In general, vulture species can obtain nearly half of their daily requirements at each feeding event (20.0–50.4%). They are adapted to ingest large amounts of food at each feeding episode and may go several days without eating. Vultures are considered obligate scavengers. They usually stay a for longer time in the surroundings of carrion (both feeding and not feeding). *A. monachus* and *G. fulvus* are considered balance scavengers (species performing a few pecks over a moderate time) and *N. percnopterus* is considered a parsimonious scavenger (small number of pecks over a very long time) [8].

Due to their diet, vultures perform an essential service by removing decomposing animal material from the environment, decreasing the risk of public health problems [9,10]. Therefore, these animals are exposed to diverse pathogenic microorganisms, some of them zoonotic, such as *Bacillus anthracis* or *Mycobacterium* spp. [1,3]. However, they have adapted to resist the microorganisms in their diet. One of their physiological adaptations against these agents is a low stomach pH level (pH 1–2) that destroys most existent microorganisms [11].

Vultures are among the largest flying birds in the world, showing high sociality and the ability to cover large areas in search of food sources [2]. For example, *G. fulvus* can forage annually in areas from 1560 to 4233 km^2^ to satisfy their food requirements [12,13].

Due to their large size, they can consume significant quantities of food at each feeding site and carry greater body reserves. Their gliding flight, where they take advantage of upward air movements, enables them to travel rapidly over long distances with relatively little energy expenditure [2]. This allows them to search for food more efficiently than other terrestrial scavengers. Avian scavengers, such as vultures, are more abundant in open landscapes and decrease as vegetation cover increases, with facultative mammal scavengers increasing in these areas. In open areas, carcasses are easily detected and consumed faster due to the presence of obligatory bird scavengers [14]. Older and bigger animals dominate and can intimidate smaller scavengers (birds and mammals) from the carcass sites [15].

The vulture population has been declining since the 19th century in Europe, with some populations already near extinction [16]. There are numerous reasons behind the vulture decline, such as poisoning (unintentional killing or deliberate poisoning through consumption of contaminated carcasses), vulnerability to bioaccumulation of toxic compounds, lead poisoning, shooting, electrocution [17], collision with power lines [18], and collision with wind turbines [2,19,20].

In the 1990s, three species of vulture endemic to South and Southeast Asia began to decline by up to 50% per year, associated with involuntary poisoning through the veterinary use of NSAIDs on cattle (diclofenac, aceclofenac, and ketoprofen). The removal of these compounds from the habitats is very challenging, but some progress has been made. In some countries, such as India, the use of diclofenac has been banned since 2006 and vulture declines there have slowed or reversed [21]. Diclofenac is still a problem in other regions, where it is still available and used both legally and illegally. In Spain, despite what has been observed in South Asian countries, the government authorized the marketing of diclofenac as a veterinary pharmaceutical for use in pigs, cattle, and horses in 2013. Studies have shown that *G. fulvus* is consuming these compound residues in livestock carcasses provided at feeding stations. The dose–response relationship has not been measured for *G. fulvus*, but experimental studies indicate that it has a similar susceptibility as *G. bengalensis*. Even though there are regulations that control veterinary treatments on carcass disposal in Spain, there is always a possibility of a contaminated carcase with diclofenac or other NSAID being supplied to vultures. This phenomenon has already been observed in a *G. fuvus* that was found dead recently in Spain with visceral gout associated with high levels of flunixin (NSAID) in the liver and kidneys. In the Spanish vulture population, it is possible that a decrease up to 0.9–7.7% per year will be seen in the future due to the use of NSAIDs [22,23].

Previous outbreaks of animal diseases, such as bovine spongiform encephalopathy, restricted the use of carcasses and animal products. According to the CE 1774/2002 Regulation, all dead farm animals had to be removed [10]. This lead to a food shortage, which has been linked to lower breeding success, higher mortality in juvenile vultures, and behaviour alterations (attacks on cattle and bigger dislocations in search of food) [24,25,26]. This scenario led to the development of the 2003/322/CE 2005/830/CE regulation that controls the use of animal by-products as food for necrophagous birds. Although these feeding stations offer food free of pathogens and toxins (i.e., reduce carcasses containing veterinary-prescribed drugs), these measures are expensive and the food resources have become artificial, with repercussions on vultures that are still unknown [26].

Loss and degradation of vulture habitats are also related to this decline due to resources and food shortages in human habitat areas [2,27]. All these factors associated with their delayed maturity and low productivity rates make their populations particularly vulnerable [28].

Even though data on the causes of mortality and morbidity of vultures have been published previously [18,19,22], long-term studies about the admissions of these animals to rehabilitation centres are scarce, particularly in Europe. Moreover, many studies only focus on one cause of death, such as electrocution or collisions with wind turbines (12). The main purpose of this study was to collect data from vulture admittance records, from one of the major wildlife rehabilitation centres located in the north of Portugal, describing admission causes, species more affected, distribution, outcomes, primary causes of death, and main lesions observed in postmortem examinations.

## 2. Materials and Methods

This study was conducted at the Wildlife Rehabilitation Centre of the University of Trás-os-Montes and Alto Douro (CRAS-UTAD) and Necropsy Service of UTAD (41_17018.13″ N–7_44021.94″ W) in northern Portugal.

We examined all vulture admittance records from CRAS-UTAD between 2005 and 2022. Only vulture species were considered. The following information was extracted from the records: species, date of admission and season (spring, summer, autumn, and winter, according to the astronomical definition in the Northern Hemisphere), rescue location (locality/city), species, age, sex, the location found, the admission cause, outcome, and date of release/death.

For each animal, the cause of admission was determined as one of the following: casual encounter (found in an exposed/vulnerable position in fields, roads, or near buildings but with no apparent lesions/disease), vehicle collision, shooting, electrocution, found debilitated, held in captivity, and unknown origin.

Definitive diagnoses or attempts at diagnosis were made during the admission process based on the physical examination and the results of additional tests, such as radiographs, haematology, cytology, and toxicological analysis, or at postmortem examination. The diseases were categorized as traumatic and non-traumatic.

The outcome of each animal was categorized as follows: euthanized, died during the recovery process, transferred to other centres, or released back into the wild.

All data collected were organized in Excel sheets, and descriptive statistics were performed.

## 3. Results

Eighty-four animals were admitted to the CRAS-UTAD and the UTAD Necropsy Service between January 2005 and December 2022, with an average of 10.5 animals/year, including live and dead animals. Of these, 10 were *A. monachus*, 69 were *G. fulvus*, and 5 were *N. percnopterus*. Unfortunately, there were no data available regarding sex and age. Regarding the age, a precise age was not available, but the majority of the animals were juveniles or young adults.

### 3.1. Geographical Origin

Figure 1 shows a map with the distribution of the admitted animals based on the location where they were collected. Most of the animals were collected from the Vila Real, Bragança, and Porto districts.

### 3.2. Annual and Seasonal Trends

Over the seventeen years studied, the most common species admitted to CRAS was *Gyps fulvus*, except in 2007 when two of the four animals were *A. monochus*. Concerning total admission numbers, 2019 was the year with the highest number of individuals (*n* = 12), as illustrated in Figure 2.

A bimodal distribution was observed for the season, with the highest vulture admission rate in summer (45.2%) and autumn (36.9%). The most admitted species in all seasons was *G. fulvus* (Figure 2). While *G. fulvus* were admitted to the centre throughout the year, this was mostly in summer and autumn. All species were admitted less often in winter (Figure 3).

### 3.3. Causes of Admission

The main causes for admission were as follows: 80% (*n* = 63) unknown cause, 13% (*n* = 10) found debilitated, 6 % (*n* = 5) vehicle collision, 4% (*n* = 3) captivity, 1% (*n* = 1) gunshot, and 1% (*n* = 1) electrocution.

Figure 4 represents the distribution of the admission causes over the different years. Transference from other centres was the main cause of admission during all the years.

Analysing the admission causes within the species of vultures (Table 1), in *G. fulvus* the main cause of admission was unknown (*n* = 50, 63.3%); in *A. monachus*, unknown cause (*n* = 10, 12.6%) had the same expression values; and in *N. percnopterus*, unknown cause was (*n* = 3, 3.8%).

The seasons with the highest number of admissions were autumn and summer for unknown causes. In autumn (*n* = 21) and winter (*n* = 2), the main cause of admission was unknown; in summer, it was electrocution (*n* = 16); and in spring, it was unknown (*n* = 5).

### 3.4. Outcome and Release Rate

Analysis of outcome data showed that 73% (*n* = 58) of the vultures that arrived at the centre could be rehabilitated and released back to the wild, while 16% (*n* = 13) died during treatment and 6% (*n* = 5) had to be euthanized due to the severity of their injuries. Releasing back into the wild was the most common outcome in all three species (Table 2).

Unknown cause was the main cause of admission in animals that were released back into the wild (*n* = 42), followed by death (*n* = 11) and being transferred to another centre (*n* = 6) (Table 3).

Considering the more precise diagnosis, the most frequent diagnosis for the animals that survived was non-traumatic (*n* = 59). Exhaustion (*n* = 11), inability to fly (*n* = 21), and dehydration (*n* = 7) were the most common signs observed. Regarding the traumatic causes of morbidity, the main cause of admission was a fracture of unknown origin (*n* = 15), followed by multiple lesions associated with collisions (vehicles, structures) (*n* = 4). For *G. fulvus*, the most frequent causes were exhaustion (*n* = 10) and pelvic limb trauma (*n* = 6). Regarding *A. monachus* (*n* = 4) and *N. percnopterus* (*n* = 2), the most common diagnosis was the inability to fly (Table 4).

A total of 18 animals died; 13 died during treatment at CRAS-UTAD and 5 were found dead and delivered to the UTAD Necropsy Service. The causes of death for these 18 animals (2 *A. monachus*, 15 *G. fulvus*, and 1 *N. percnopterus*) were traumatic (*n* = 9), non-traumatic (*n* = 6), and unknown (*n* = 3). Nine animals died due to traumatic origin: two due trauma of unknown origin, one cervical fracture, two gunshots, and two electrocution. Of these, six had infectious diseases (*n* = 3) and one was emaciated and dehydrated. In three cases, the diagnosis was prejudiced due to the corpse’s decomposition (Table 4). In Figure 5, Figure 6, Figure 7, Figure 8 and Figure 9, there are some examples of the lesions observed in the postmortem exam.

Regarding *G. fulvus*, the most common cause of death was non-traumatic with three animals with aspergillosis, followed by unknown (*n* = 3). In *A. monachus*, only two animals died, one due to electrocution and the other euthanised due to metabolic disease and poxvirus. The only *N. percnopterus* that died was due to a gunshot (Table 5).

## 4. Discussion

Vultures are a critical part of ecosystems, and population declines worldwide have been linked to anthropogenic causes, such as toxins or collisions [19]. In the Pyrenean bearded vulture (*Gypaetus barbatus*) population, high levels of lead were detected in the liver and kidney of two individuals (from a sample of 130), suggesting the risk of lead poisoning in these animals [22].

The present study was the first to investigate admission causes, outcomes, morbidity, and mortality of vultures admitted to a wildlife rehabilitation centre in Portugal. Studies concerning vultures’ mortality are very scarce in Europe [15,18,30], and in Portugal, they are almost non-existent.

Our data were drawn from the records of one of the largest wildlife rehabilitation centres in Northern Portugal—CRAS-UTAD, over 17 years (2005–2022). Furthermore, the injured animals were received from different social organizations (Serviço de Proteção da Natureza e do Ambiente (SEPNA), police, ICNF, etc.) and directly from unknown citizens. Also, animals that were found dead in the wild were delivered to the UTAD necropsy service to determine the cause of death.

Eighty-four vultures were admitted to both services during this period, with most of the animals being collected in the regions near the centre, as observed in Figure 1. This number seems to be not very high. However, it is important to note that two species are endangered, with fewer individuals in the territory. There are only 40 breeding couples of *A. monachus* nesting in Portugal at the moment. Regarding the other two species, information is still not available. The most common species admitted to CRAS was *G. fulvus* (69 animals); the most common species in Portugal and the least endangered (4).

Concerning the distribution of species in Portuguese territory, most of the animals admitted were *G. fulvus*, since it is the species with the widest distribution in Portugal throughout the entire territory. *N. percnopterus* was the only species that was collected in areas where it is not common to be present, namely the north of Portugal [4].

One of the study’s limitations was the lack of data on sex and age. In these animals, sex can only be obtained by molecular techniques (PCR) or endoscopy [31].

Summer (45.2%) and autumn (36.9%) were the main admission seasons. The nesting period in Portugal for these species is between March and August [32]; therefore, in the summer period, it is normal to have a higher number of juveniles that are more susceptible to collisions or debilitation due to the lack of food resources [33]. Furthermore, these seasons are also periods of migrations, particularly for species such as *N. percnopterus* [34]. Therefore, during this period, a higher number of these animals are expected in the territory, and animals can be debilitated after long migrations [25].

In our study, the main cause for admission was 80% (*n* = 63) unknown causes. Unfortunately, there are not much data regarding admission causes in vultures since most articles reference only mortality [19,25].

The main cause of morbidity of the animals in our study was non-traumatic (*n* = 59). Exhaustion (*n* = 11), inability to fly (*n* = 21), and dehydration (*n* = 7) were the most common signs observed. Regarding the traumatic causes of morbidity, the main cause of admission was a fracture of unknown origin (*n* = 15), followed by multiple lesions associated with collisions (vehicles, structures) (*n* = 4). In a review performed by Ives et al. (2022), they confirmed that the main cause of morbidity is due to non-traumatic causes, such as toxins (lead, pesticides, and non-steroidal anti-inflammatory drugs (NSAIDs) [19]. Although there was a suspicion in our study, it was impossible to confirm. However, a higher number of animals may be underdiagnosed, as a definitive diagnosis was not achieved. Therefore, we cannot exclude that AED of animals may not have a toxic cause. In Portugal, there has been a concern, namely with the antidote project. In the last ten years, 1534 cases of poisoned animals (domestic and wild) were identified in the country, with *G. fulvus* being one of the most affected specie [35]. In Spain between 2000–2010, 8000 cases of illegal poisoning were documented, including a critical number of *N. percnopterus*, *A. monachus*, and *G. fulvus* [36].

The second cause of morbidity in this study was trauma, also observed in other studies, such as Ives et al. (2022) [19]. Regarding traumatic injury, most seemed to be associated with anthropogenic origins such as collision, electrocution, or gunshot. Margalida et al. (2008) [26] also confirmed that the main causes of mortality in the European population of bearded vultures were related to anthropogenic factors [25]. In the present study, except for the animals that were underweight and too weak to fly, most of the animals were admitted due to traumatic injuries: multiple fractures (*n* = 18), collision (*n* = 5), electrocution (*n* = 2), and gunshot (*n* = 3). These traumas could be secondarily related to poisoning, since poisoned animals are more debilitated and can easily become victims of collisions or being run over [37,38].

The number of admissions in the last few years has been increasing. This phenomenon can be associated with a better understanding of the population regarding the conservation of these animals, and to sanitary regulations affecting the number of individuals found in the field (2006–2011). Infected cattle with bovine spongiform encephalopathy (BSE) lead to the imposition of sanitary legislation (Regulation CE 1774/2002) that significantly restricted the use of animal-origin products that were not intended for human consumption. Most of the cattle carcasses were destroyed and supplementary feeding points for necrophages were greatly diminished (−80%) because of those sanitary regulations. The lack of food resources could have contributed to the increase in vulture submissions in WRC [36].

Concerning mortality, only 13 animals died during treatment and 5 were found dead. Of these, 3 were due to infectious diseases, 10 were due to traumatic injuries, and 3 were too decomposed to determine the cause of death.

In our data, only one animal death was associated with intoxication. There may have been more, but due to the lack of resources it was not possible to confirm these cases. Infectious disease in vulture health is the most neglected area of research. The most common infectious diseases are due to *Aspergillus fumigatus*, an opportunistic fungal pathogen. This agent was observed in three animals that ended up dying. Another three animals presented infectious diseases, such as West Nile Virus (*n* = 1), poxvirus (*n* = 1), and bacterial infection (*n* = 1).

Regarding the outcomes of the animals admitted to CRAS-UTAD, 73% (*n* = 58) were released back into the wild. As happens with admissions, data on outcomes are almost non-existent and unavailable. CRAS-UTAD has a rehabilitation centre and a satellite centre where animals can recover to be released back into the wild. Most of the animals were kept only for a short period in the centres to recover and be released since the admissions corresponded to healthy animals that were found to be accidentally slightly underweight, dehydrated, and too weak to fly. However, in cases of fractures and more severe lesions, the animals remain hospitalized for long periods and require medical care and rehabilitation to be released back to their natural habitat.

## 5. Conclusions

In conclusion, of the 84 animals admitted to wildlife rehabilitation centres in northern Portugal, more than half recovered and were released back into the wild. The main causes of admission were related to random findings and debilitating physical conditions. The main causes of mortality were associated with trauma, and in most cases linked to anthropogenic sources.

In the future, more funding for wildlife health, including necropsy, toxicology services, and public databases for causes of vulture mortality and morbidity, is necessary. Monitorization of the animals after release (e.g., GPS) should be performed to understand if the animals are adapting to their habitat and identify behaviour alterations associated with anthropogenic causes. Also, the public needs to become more involved in the activities of wildlife rehabilitation centres and similar environmental associations, demystifying the bad reputation of vultures.

## Figures and Tables

**Figure 1 animals-13-02093-f001:**
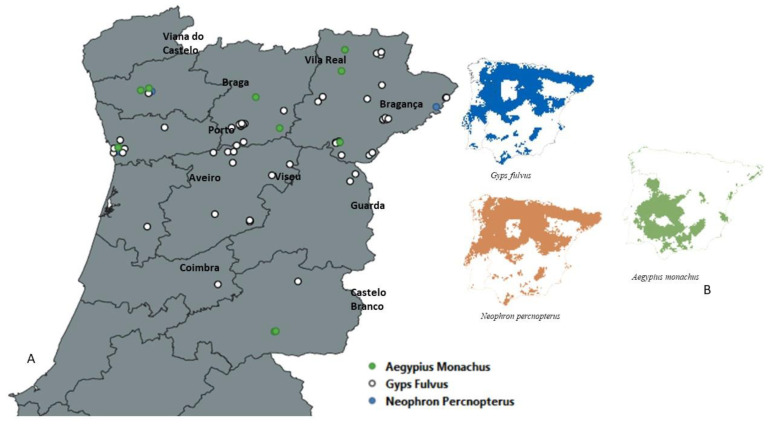
(**A**) Spatial distribution of the animals admitted from 2005–2022 in the Wildlife Rehabilitation Centre of the University of Trás-os-Montes and Alto Douro (Vila Real, Portugal). (**B**) Spatial distribution in the Iberian Peninsula of *A. monachus, G. fulvus,* and *N. percnopterus*, adapted from [29].

**Figure 2 animals-13-02093-f002:**
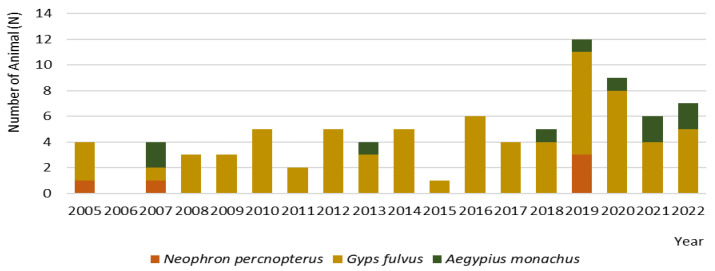
Number of animals admitted from 2005–2022 in the Wildlife Rehabilitation Centre of the University of Trás-os-Montes and Alto Douro (Vila Real, Portugal).

**Figure 3 animals-13-02093-f003:**
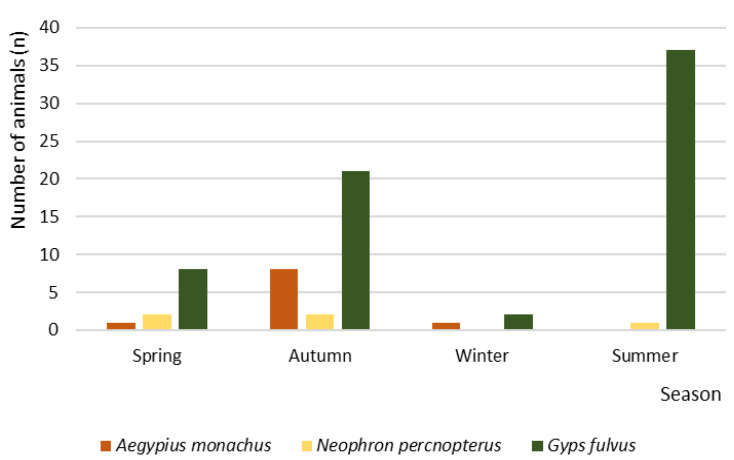
Number of animals admitted in each season (winter, summer, spring, autumn) from 2005–2022 in the Wildlife Rehabilitation Centre of the University of Trás-os-Montes and Alto Douro (Vila Real, Portugal).

**Figure 4 animals-13-02093-f004:**
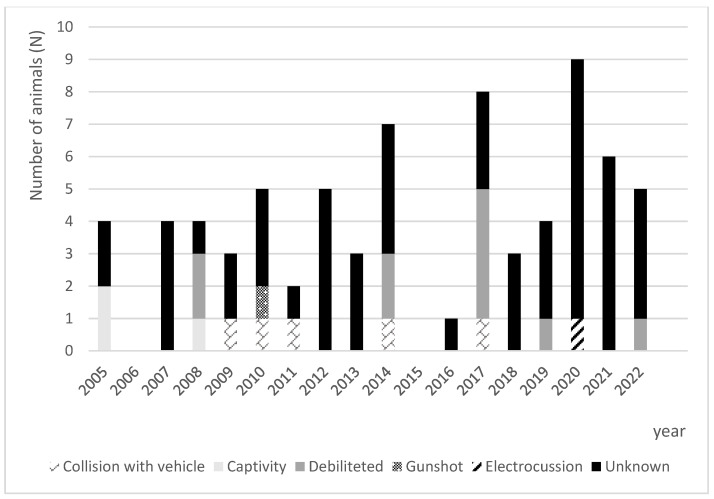
Causes of admission to the Wildlife Rehabilitation Centre from the University of Tras-os-Montes and Alto Douro (Vila Real, Portugal) from 2005 to 2022.

**Figure 5 animals-13-02093-f005:**
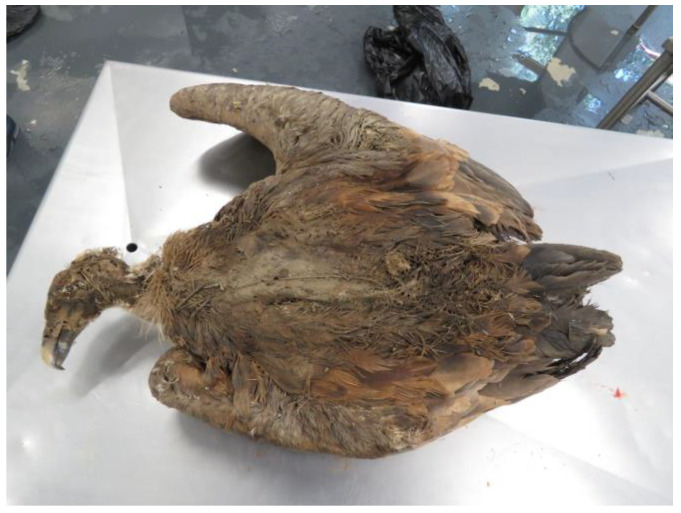
*Gyps fulvus* in an advanced state of decomposition.

**Figure 6 animals-13-02093-f006:**
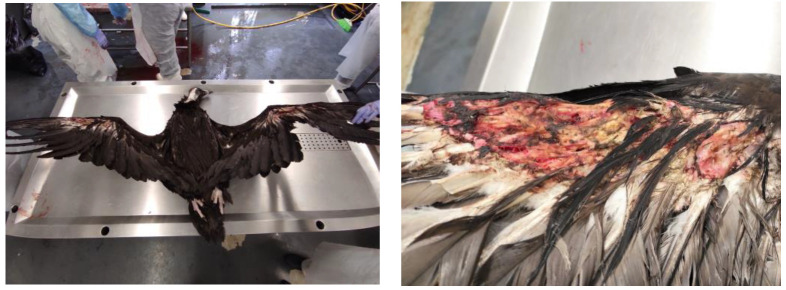
Electrocussion in an *Aegypius monachus*.

**Figure 7 animals-13-02093-f007:**
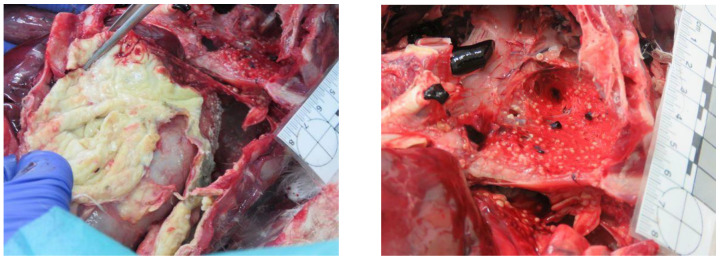
Avian aspergillosis in a *Gyps fulvus*.

**Figure 8 animals-13-02093-f008:**
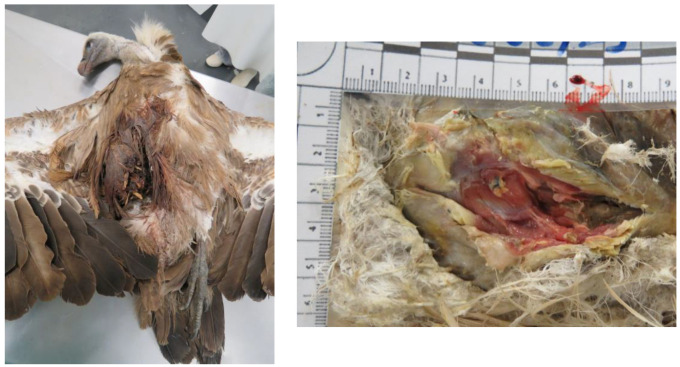
Gunshot wound and projectile in *Gyps fulvus* (**right**) and *Neophron percnopterus* (**left**).

**Figure 9 animals-13-02093-f009:**
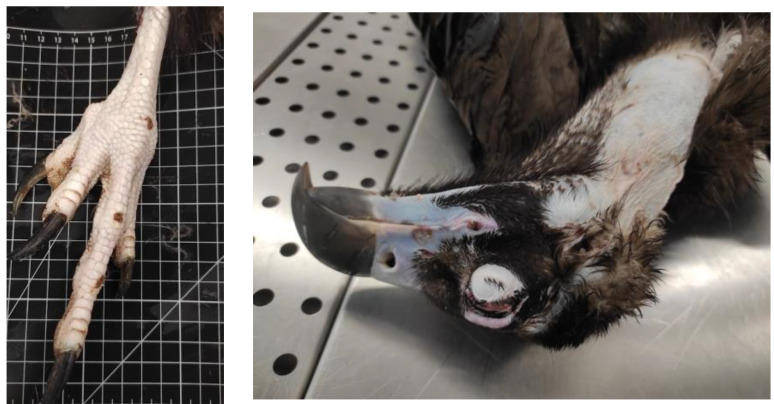
Pox virus lesions in an *Aegypius monachus*.

**Table 1 animals-13-02093-t001:** Frequency and percentage of the causes of admission concerning season, specie, and outcome.

	Vehicle Collision	Captivity	Debilitated	Gunshot	Electrocution	Unknown
**Species**						
*Gyps fulvus*	6 (7.5)	2 (2.5)	9 (11.3)	1 (1.9)	1 (1.3)	50 (63.3)
*Aegypius monachus*	0 (0)	1 (1.3)	1 (1.3)	0 (0)	0 (0)	10 (12.6)
*Neophron percnopterus*	0 (0)	1 (1.3)	0 (0)	0 (0)	0 (0)	3 (3.8)
Total	6	4	10	1	1	63
**Season**						
Winter	0 (0)	0 (0)	0 (0)	0 (0)	0 (0)	3 (3.8)
Autumn	2 (2.5)	1 (1.3)	2 (2.5)	0 (0)	0 (0)	29 (36.8)
Spring	0 (0)	2 (2.5)	0 (0)	0 (0)	0 (0)	5 (6.3)
Summer	3 (3.8)	0 (0)	8 (10.1)	1 (1.9)	1 (1.3)	22 (27.8)
Total	5	3	10	1	1	59

**Table 2 animals-13-02093-t002:** Frequency and percentage of the outcomes concerning specie.

	*Gyps fulvus*	*Aegypius monachus*	*Neophron percnopterus*
Released into the wild	50 (63.3%)	5 (6.3%)	3 (3.7%)
Death	15 (19%)	2 (2.5%)	1 (1.63%)
Non/euthanasia	5 (6.3%)	0 (0%)	0 (0%)
Euthanasia	10 (12.7%)	2 (2.5%)	1 (1.3%)
Transference to another centre	4 (5.1%)	3 (3.7%)	1 (1.3%)

**Table 3 animals-13-02093-t003:** Frequency and percentage of the causes of admission concerning the outcome.

	Vehicle Collision	Captivity	Debilitated	Gunshot	Electrocution	Unknown
**Outcomes**						
Released into the wild	5 (6.3)	3 (3.7)	8 (10.1)	1 (1.3)	0 (0)	42 (53.1)
Death	0 (0)	0 (0)	1 (1.3)	0 (0)	1 (1.3)	11 (13.9)
Transference to another centre	0 (0)	0 (0)	4 (5.1)	0 (0)	0 (0)	6 (7.6)

**Table 4 animals-13-02093-t004:** Traumatic and non-traumatic causes of disease in vultures admitted to the Wildlife Rehabilitation Centre from the University of Tras-os-Montes and Alto Douro (Vila Real, Portugal) from 2005 to 2022. Data from the animals that were released back to the wild or transferred to another Centre.

	*Gyps fulvus*	*Aegypius monachus*	*Neophron percnopterus*
**Traumatic causes of disease**			
Multiple trauma	4		1
Gunshot	1		
Pelvic limb trauma	6		
Extensive muscle necrosis	1	1	
Thoracic limb trauma	6	1	
**Non-traumatic causes of disease**			
Toxic	1		
Ascites	1		
Exhaustion	10	1	
Enteritis of unknown origin	1		
Inability to fly	15	4	2
Dehydration	7		
Malnutrition	5		
Convulsion of unknown origin	1	1	
Ataxy	1		
Corneal opacity			1
*Infectious*/*parasitic*			
Feather Lice	2		
West Nile Virus	1		
Bacterial infection	1		
*Ascaris* spp.	1		
*Leucocytozoon* spp.	1		

**Table 5 animals-13-02093-t005:** Mortality caused by vultures admitted to the Wildlife Rehabilitation Centre from the University of Tras-os-Montes and Alto Douro (Vila Real, Portugal) from 2005 to 2022.

	*Gyps fulvus*	*Aegypius monachus*	*Neophron percnopterus*
**Traumatic causes**			
Trauma unknown origin	2		
Cervical fracture	1		
Thoracic limb trauma	2		
Gunshot	1		1
Electrocution	1	1	
**Non-traumatic causes**			
Emaciated and dehydrated	1		
Osteomyelitis and muscle necrosis	1		
Poxvirus and metabolic disease		1	
Aspergillosis	3		
**Unknown (decomposition)**	3		

## Data Availability

Not applicable.

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
