# Peer review of "Admission Causes, Morbidity, and Outcomes in Scavenger Birds in the North of Portugal (2005–2022)"

_animals, 2023, doi:10.3390/ani13132093_

Round 1
Reviewer 1 Report
I think the contribution is interesting but limited because the number of cases are reduced and the main factors discussed very general. However, with a more deep discussion in relation to the management and conservation of the vulture species (diclofenac, attacks, illegal poisoning, sanitary policies) the manuscript could increase the scope and interest. The current version have very narrow and general descriptive parts that could be improved with the abundant literature available. I have suggested some references to help the authors and I have added all my comments directly in the PDF.

In my opinion the English could be improved.
Author Response
# Reviewer 1
I think the contribution is interesting but limited because the number of cases are reduced and the main factors discussed are very general. However, with a deeper discussion in relation to the management and conservation of the vulture species (diclofenac, attacks, illegal poisoning, sanitary policies) the manuscript could increase the scope and interest. The current version has very narrow and general descriptive parts that could be improved with the abundant literature available. I have suggested some references to help the authors and I have added all my comments directly in the PDF.
Authors answer: thank you for all the suggestions and comments on the manuscript. We try to answer all the questions. All corrections are marked in yellow.
1 - but this is not an admission cause, you should quote here the cause of admission in the other centre or add this to the unknown cause.
Authors answer: the authors agreed, this admission cause has been deleted and added to the unknown cause.
2 - This is useful but a very general and descriptive information
Authors’ answer: the authors added more information to this section. “The 16 species of Accipitridae can be found in Africa, Asia, and Europe, and 7 species of Cathartidae can be found in North and South America. (2,3).”
3 - You could quote more details about the status and distribution. For example, the importance of the Iberian Peninsula from a European context.
Authors’ answer: the authors added more information to this section regarding the situation in Portugal and in Europe.
4 - Very general sentence. Here you could discuss differences in dietary habits among obligate and facultative species (see for example Moreno-Opo et al. 2016 Behav Ecol).
Authors’ answer: the authors added more information to this section.
5- About foraging ecology I suggest revising and quote (if necessary) papers such as Delgado-González et al. 2022 Sci Rep, Morant et al. 2023 Ecol Evol and references quoted. About the vulture efficiency see Sebastián-González et al. 2020 Ecograpy or Oliva-Vidal et al 2022 Oikos. For hierarchies and dominance see Moreno-Opo et al. 2020 Sci Rep or Moreno-Opo et al 2016 Behav Ecol
Authors’ answer: the authors added more information to this section.
6 - But the European population is the only with recovery. See Safford et al 2019 about conservation problems on Old World vultures. I miss also references about the impact of poisoning on Egyptian and cinerous vultures (see Hernández & Margalida 2007 Eur J Wildl Res, 2008 Ecotoxicology), diclofenac (Green et al 2016 J Appl Ecol).
Authors’ answer: the authors added more information to this section.
7 - You can also quote specific papers about the demographic impact of sanitary regulations on vultures. For example Almaraz et al 2022 Ecol Appl, margalida et al 2014 Ecol Appl
Authors’ answer: the authors added more information to this section.
8 - Please add references supporting this sentence
Authors’ answer: references added.
9 - If possible, an interesting information should add the distribution of the three vulture species with the spatial distribution of the individuals admitted
Authors’ answer: information added.
10 - I'm not sure about this category. 23% from other centres is not a cause of admission. This should be added to unknown cause if is not possible to identify the cause of admission on the first centre.
Authors’ answer: the category was eliminated and added to unknown.
11 - According to the figure the last year increased the number of individuals. You should discuss if this is cause of an increase or a more important effort in the field.
Authors’ answer: figure 3 had a mistake, that was corrected. The last year was not the year with more submissions. Submissions have increased a little during the years but there were years with very few submissions.
12 - please could you add more details about the age of the individuals? Perhaps younger individuals are prone to exhaustion, inability to fly...
Authors’ answer: we do not know the age of the animals, that information is leaking from the files. But yes, most animals were juveniles and young adults. That information was added to the results.
13 - Please discuss this spatial pattern with the distribution of the species in the country
Authors’ answer: information added.
14 - This sentence "In addition... these exams" could be removed
Authors’ answer: removed.
15 – and what is the percentage of vultures affected?
Authors’ answer: the data is not available.
16 - Here I miss if sanitary regulations affected the number of individuals found in the field (2006-2011). Also, if the emergent conflict between vultures and livestock could affect the admissions of individuals during the last years (see the cases in Spain with problems related with poisoning, Margalida et al. 2011 Nature
Authors’ answer: the information was added.
Reviewer 2 Report
The article provides a comprehensive analysis of the conservation importance of vultures. It is highly recommended to search for a statistical relationship between the investigated factors and reasons for admission, as it would greatly assist in planning future conservation activities. This approach would provede valuable insights, enabling targeted efforts to address threats and maximize conservation impact for vulteres. This enhanced approach would futher contribute to heihtened scientific validity.
Author Response
# Reviewer 2
The article provides a comprehensive analysis of the conservation importance of vultures. It is highly recommended to search for a statistical relationship between the investigated factors and reasons for admission, as it would greatly assist in planning future conservation activities. This approach would provede valuable insights, enabling targeted efforts to address threats and maximize conservation impact for vulteres. This enhanced approach would futher contribute to heihtened scientific validity.
Authors’ answer: thank you for the suggestions, we added more information to the manuscript to improve it. Due to the reduced number of animals and the lack of some information, a statistical relationship would not be very useful. It could lead to some bias in the author's opinion.
Reviewer 3 Report
The authors present a summary of different data (mortality, morbidity, admission and outcomes) of three vulture species in a Portuguese rescue centre. Although not many definitive and structured conclusions can be taken from most of the presented data, I believe these preliminary data are important to be published in order to pan future studies or conservation projects. Therefore, I believe these authors should receive credit for their work.
Here I present some comments/suggestions that I believe are important to be addressed:
- L16-19; 28-30; 143-145: I am not sure if "transference from other rescue centres" should be considered a cause of admission like, for instance, "vehicle collision", mainly because an animal could be transferred from a rescue centre to other due to better facilities to manage recovery from a specific reason (as collision). Therefore, I am not sure if "transfer from other rescue centres" should not be eliminated as a category and distributed among the others. This way it would provide a better idea of the different relevance of each reason for admission in vultures.
- L104-106: I could not find the information on the normality test results. Which normality test did you perform?
- From my perspective, there is no reason to have Figure 4, since you provide similar information in table 1, so it is a repetition.
- L133-134: this sentence should be rephrased because both species share Autumn as a season, so there is no contrast that justifies the use of these connectors.
- Some Latin names of species are not in italic (L 234, L 275...).
- L232-233: A reference for this sentence should be provided.
I have nothing else to add and I wish the authors everything good.
Author Response
Reviewerer 3
The authors present a summary of different data (mortality, morbidity, admission and outcomes) of three vulture species in a Portuguese rescue centre. Although not many definitive and structured conclusions can be taken from most of the presented data, I believe these preliminary data are important to be published in order to pan future studies or conservation projects. Therefore, I believe these authors should receive credit for their work.
Authors’ answer: thank you for the suggestions.
Here I present some comments/suggestions that I believe are important to be addressed:
- L16-19; 28-30; 143-145: I am not sure if "transference from other rescue centres" should be considered a cause of admission like, for instance, "vehicle collision", mainly because an animal could be transferred from a rescue centre to other due to better facilities to manage recovery from a specific reason (as collision). Therefore, I am not sure if "transfer from other rescue centres" should not be eliminated as a category and distributed among the others. This way it would provide a better idea of the different relevance of each reason for admission in vultures.
Authors’ answer: this category was eliminated and added to the category “Unknown”
- L104-106: I could not find the information on the normality test results. Which normality test did you perform?
Authors’ answer: it was a mistake. Due to the very low number of animals and the missing information in some fields such as sex and age, the testes were not performed. That sentence was removed.
- From my perspective, there is no reason to have Figure 4, since you provide similar information in table 1, so it is a repetition.
Authors’ answer: the authors think that both are necessary since Figure 4 gives information regarding the years and in the table regarding specie end seasons and there are some variations between them even if some points are very similar.
- L133-134: this sentence should be rephrased because both species share Autumn as a season, so there is no contrast that justifies the use of these connectors.
Authors’ answer: sentence corrected.
- Some Latin names of species are not in italic (L 234, L 275...).
Authors’ answer: corrected.
- L232-233: A reference for this sentence should be provided.
Authors’ answer: sentence corrected.
Round 2
Reviewer 1 Report
Thanks for your effort. In my opinion, the piece have been improved substantially. I have added some comments to improve the current version. Please take attention to quote correctly the references to support different sentences. Please see my comments in the PDF version.

Author Response
Dear Editor,
Thank you for the comments and suggestions. All have been added in the manuscript.
To answer some questions that were nor possible to add to the manuscript:
1 - Please could you update the census? The information refers to 23 years ago and probably the current status is different
Authors answer: There is no current census. The new census only started this year and only be available in the end on year or the beginning of the next, in all three species.
2 - Why is not possible? The necropsies do not provided enough details?
The author answers that the poisoning-associated lesions are very general - haemorrhoids, pale muscles, and hematomas... These lesions can be associated with other pathologies. Only the necropsy cannot confirm if it is poisoning, only in the cases where the bait is found inside the stomach. It is always necessary to perform toxicological exams. In our case, there was no money available to perform these exams (they were very expensive).
